# Size, shape, and direction matters: Matching secondary genital structures in male and female mites using multiple microscopy techniques and 3D modeling

Orlando Cómbita-Heredia[1,2]* , Connor J. Gulbronson[3¤], Ronald Ochoa[4], Edwin Javier Quintero-Gutiérrez[5], Gary Bauchan[4,6†], Hans Klompen[1]

**1** Acarology Laboratory, Ecology Evolution and Organismal Biology EEOB, Ohio State University, Columbus, Ohio, United States of America, **2** Centro de Investigación en Acarología, Bogotá, Colombia, **3** Oak Ridge Institute for Science and Education (ORISE) Postdoctoral Fellow, Floral and Nursery Plant Research Unit, United States Department of Agriculture, Agricultural Research Service, U.S. National Arboretum, Beltsville, Maryland, United States of America, **4** Agricultural Research Service, Systematic Entomology Laboratory, United States Department of Agriculture, Beltsville, Maryland, United States of America, **5** Instituto Colombiano Agropecuario (ICA), Manizales, Colombia, **6** Agricultural Research Service, Soybean Genomics and Enhancement Laboratory, Electron and Confocal Microscopy Unit, United States Department of Agriculture, Beltsville, Maryland, United States of America

☉ These authors contributed equally to this work.
† Deceased.
¤ Current address: Indiana Center for Biological Microscopy, Indiana University, Indianapolis, Indiana, United States of America
* combitaheredia.1@osu.edu

## Abstract

Studies of female genital structures have generally lagged behind comparable studies of male genitalia, in part because of an assumption of a lower level of variability, but also because internal genitalia are much more difficult to study. Using multiple microscopy techniques, including video stereomicroscopy, fluorescence microscopy, low-temperature scanning electron microscopy (LT-SEM), and confocal laser scanning microscopy (CLSM) we examined whether the complex sperm transfer structures in males of *Megalolaelaps colossus* (Acari: Mesostigmata) are matched by similarly complex internal structures in the female. While both LT-SEM and CLSM are well suited for obtaining high-quality surface images, CLSM also proved to be a valuable technique for observing internal anatomical structures. The long and coiled sperm transfer organ on the chelicera of the males (spermatodactyl) largely matches an equally complex, but internal, spiral structure in the females in shape, size, and direction. This result strongly suggests some form of genital coevolution. A hypothesis of sexual conflict appears to provide the best fit for all available data (morphology and life history).

**Data Availability Statement:** All figures and 3D models are available at http://morphobank.org/permalink/?P3717.

**Funding:** This work was supported by National Science Foundation (HK. DEB 2017439) and by Colciencias, Colombia, Doctorados en el Exterior Grant (568-2012- C.C. 80048516). Some images presented in this manuscript were generated using the instruments and services at the Campus Microscopy and Imaging Facility. This facility is supported in part by grant P30 CA016058, National Cancer Institute, Bethesda, MD. The funders had no role in study design, data collection and analysis, decision to publish, or preparation of the manuscript.

**Competing interests:** The authors have declared that no competing interests exist.

# 1 Introduction

Sexual dimorphism and genitalic specialization in mites can be very limited, but some species display quite spectacular modifications. Although such modifications are uncommon in the order Mesostigmata, most males in the infraorders Dermanyssina and Heterozerconina (Parasitiformes: Mesostigmata) have a specialized structure (the spermatodactyl) on their chelicera which acts as a sperm transfer organ (gonopod) [1]. Corresponding with this, females of Dermanyssina and Heterozerconina (usually) have a modified insemination system including a pair of secondary genital openings (solenostomes) often located near coxae III or IV [2, 3]. This system of sperm transfer (podospermy) differs from the presumed primitive system (tocospermy) in Mesostigmata where the male uses unmodified chelicera to transfer a sperm packet directly to the primary genital opening (ovipore) of the female [1]. The spermatodactyl is relatively simple in many Dermanyssina, but in some species it has become quite complex. The current study involves one of these species, *Megalolaelaps colossus* Cómbita-Heredia and Quintero-Gutiérrez (Dermanyssina: Megalolaelapidae). It presents a strikingly complex spermatodactyl (Fig 1) [4], highly coiled, resembling a corkscrew that if uncoiled could be half as long as the body (Fig 2A–2C).

The extravagant shape of the male spermatodactyl in *M. colossus* raises the question whether and how this shape is matched in the females. In general, few studies have focused on female genitalic structures, in part because of the assumption that female genital structures are less variable [5], and thus of less interest, but also because of the practical problem that female genitalia are usually internal and therefore harder to analyze. Several studies on internal reproductive structures of podospermic female mesostigmatid mites have led to the distinction of two general systems, the laelapid and phytoseiid types [3, 6–9] but these studies focused on the deep internal structures. Other studies have led to a better understanding of the configuration and development of the spermatodactyl in males [10–13]. However, males and females have rarely been studied simultaneously with the explicit goal of examining potential coevolution [5, 14]. One possible exception is a study on *Veigaia paradoxa* Willmann, where the authors note that the spermatodactyl is as long as the internal "spiral organ" in the female, supporting Willmann's hypothesis that the spermatodactyl may be inserted in this "spiral organ" [15]. Notably, comparative studies of male and female genitalic structures are required to assess the possible role of sexual selection in molding observed genitalic structures [16] and may suggest mechanisms of sexual selection such as sperm competition, cryptic female choice or sexual conflict [14, 17]. Secondary genital structures like spermatodactyl, secondary insemination systems, and associated structures fall under the category of genital traits, because they are directly involved in copulation [16].

One significant technical challenge in examining the interaction of male and female genital structures is that female reproductive structures are internal, and thus not amenable for study using regular miscroscopy techniques. Most studies of female genital structures in Mesostigmata are limited to species with fairly thin cuticles and minimal ornamentation, such as Phytoseiidae, that are studied by light microscopy of slide-mounted specimens. This approach is quite difficult with more sclerotized species. Moreover, the often clearly 3-dimensional reproductive structures are also difficult to study using standard light microscopy. Scanning Electron Microscopy (SEM) and its modifications of Low Temperature Scanning Electron Microscopy (LT-SEM) have revolutionized imaging of mite morphology, providing both high resolution and 3D images, but these techniques scan surfaces, and are therefore not useful for internal structures. Transmission Electron Microscopy (TEM) has been used to great effect in previous studies and produces highly detailed images, but this type of analysis is very time consuming and destructive. Confocal Laser Scanning Microscopy (CLSM) is a technique that

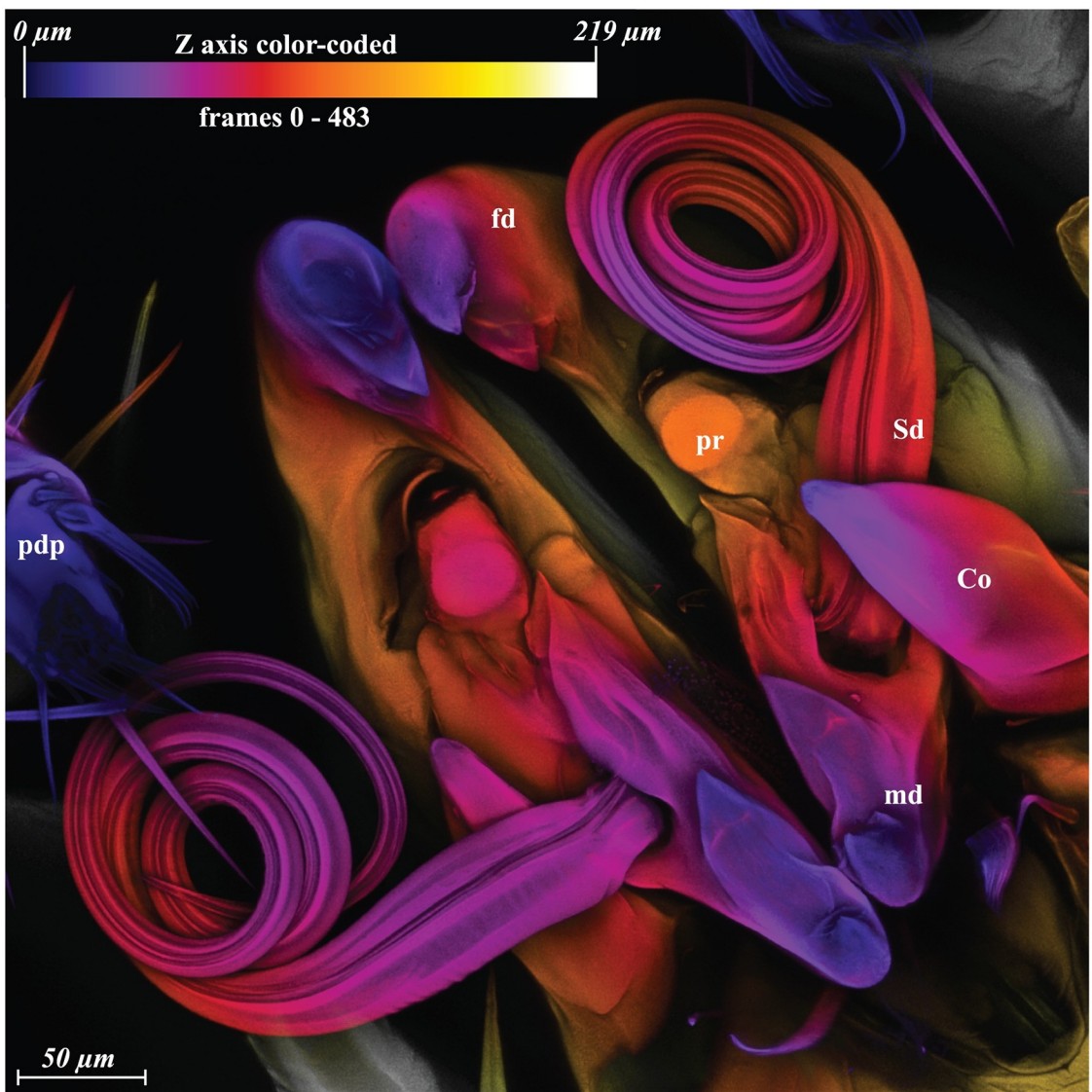

**Fig 1.** *Megalolaelaps colossus* **male, maximum intensity projection (MIP) of CLSM image of the spermatodactyl with color coded z axis.** Co, corniculi; fd, fixed digit; md, movable digit; pdp, pedipalp; pr, putative reservoir; Sd, spermatodactyl.

cannot match the resolution of TEM, but it is fast, non-destructive, can be applied to fluid and slide preserved specimens, and can help reveal 3-dimensional internal and external structures. Our initial assumption, based on previous studies done in mites of the superorder Acariformes [18–26], was that CLMS would be a successful technique for external and internal morphology, even though mesostigmatid mites have a thicker and more sclerotized cuticle than most Acariformes [27].

The primary goals of this paper are to use multiple microscopy technologies 1) for a comparative study of the internal and external structure of the secondary genital systems of male and female *Megalolaelaps colossus*, and 2) to demonstrate the uses of CLSM in investigations of anatomy of Mesostigmata. The results are used to generate a preliminary hypothesis on the role of genital coevolution in this group of mites.

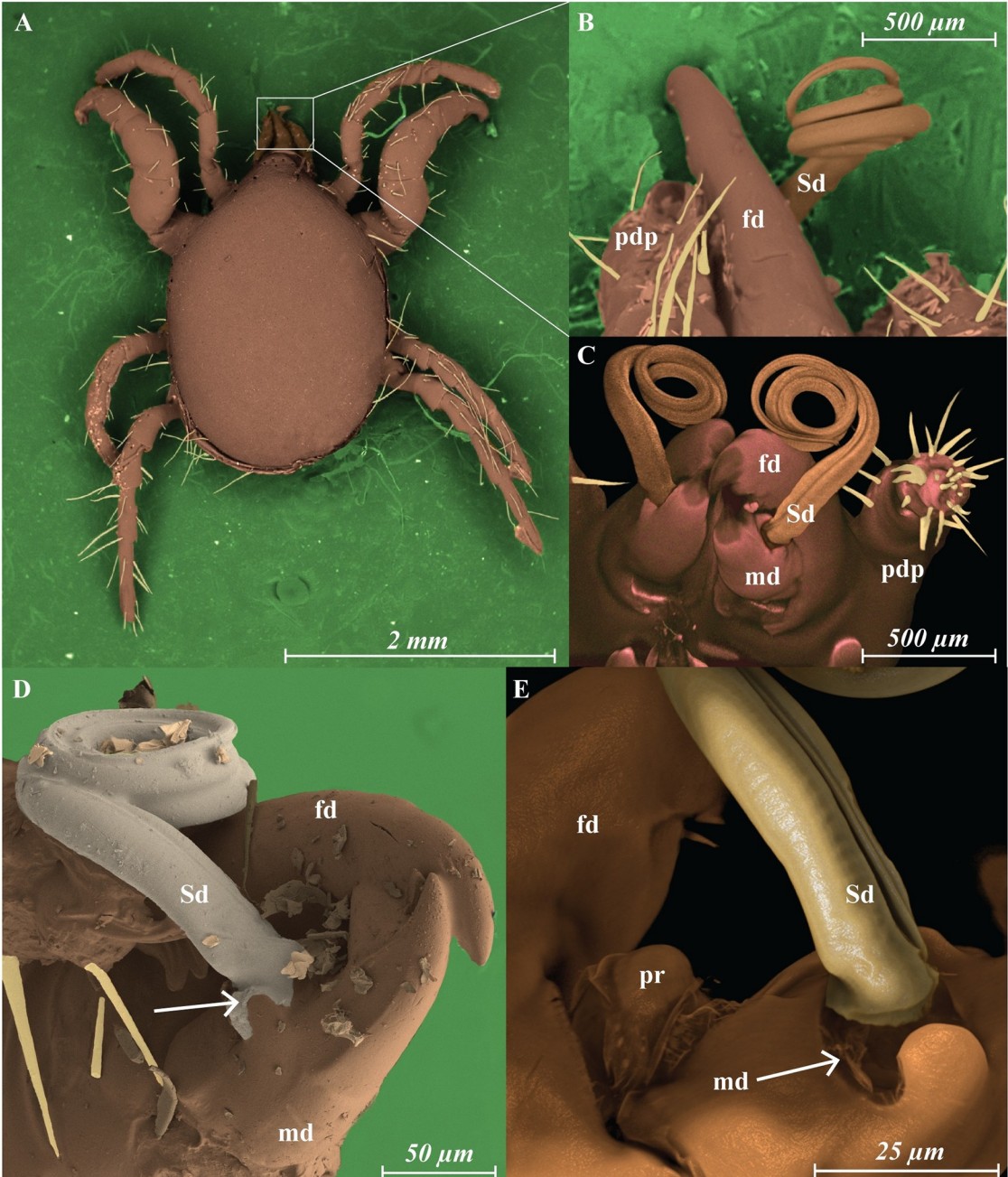

**Fig 2.** *Megalolaelaps colossus* **male (A) dorsal view LT-SEM colorized image (B) lateral view of the spermatodactyl (C) frontal view of spermatodactyl CLSM colored image (D) colorized LT-SEM (E) colorized CLSM.** Arrows: articulation between spermatodactyl and movable digit image. fd, fixed digit; md, movable digit; pr, putative reservoir; pdp, pedipalp; Sd, spermatodactyl.

## 2 Results

### 2.1 Male and female reproductive structures

Males: Previous studies of the gnathosomal structure of *Megalolaelaps* mention the spermatodactyl in males but provides few details [28]. As in all Dermanyssina, the spermatodactyl in

inserted antiaxial on the movable digit of the chelicera. It is quite long (ca. 1200 $\mu$ extended (N = 11; 1112—1285)) and appears to be movable. The left spermatodactyl is coiled counter-clockwise and the right is coiled clockwise from proximal to distal end. Based on orthogonal views of the 3D images the spermatodactyl seems to be an evagination of the procuticule of the movable digit of the chelicera. The joint between the spermatodactyl and the movable digit (Fig 3A and 3B) appears to be flexible with an expansion or "head" at the base of the spermato-dactyl inserted in a big concave area on the movable digit resembling a synovial saddle joint in vertebrates. Presumably this allows considerable twisting of the spermatodactyl relative to the movable digit. The joint is covered by a thin membrane that's visible under SEM (Fig 2D arrow) but less conspicuous with CLSM (Fig 2E arrow). The entire coiled spermatodactyl fits completely in the gnathosomal cavity or camerostome when the chelicerae are retracted.

The sperm delivery system in *M. colossus* starts in the movable digit proper as a duct that connects to a proximal process or "putative reservoir" [13]. After leaving the movable digit and extending into the spermatodactyl, the duct becomes an external groove (Fig 3A arrows). This system differs from the spermatodactyl in other podospermous species which have an internal sperm duct rather than an external groove [10, 11, 13]. The spermatodactyl in *Megalolaelaps* contains two internal cavities that appear connected to the movable digit and run on either side of the external groove). The cuticle of the spermatodactyl presents a series of coiled rugosities along the internal side (Fig 3B and 3C arrows) resembling the taenidia of trachea or perhaps more appropriately the annuli-like structures present on the cheliceral shaft of some Uropodina (Mesostigmata), e.g. *Uroactinia* [29]. We assume that these structures permit shape modifica-tion, specifically uncoiling, of the spermatodactyl. Additionally, we observed inside each cavity a structure we could not identify that could be unfolded (Fig 3D) or folded (Fig 3E).

Females: The secondary genital openings are located on coxae III (Fig 4A, 4B, 4D and 5 arrows). Internally, they open into a cavity in the coxa that may function as a sperm pocket (Fig 4B). Subsequently, and in the body proper, they connect to a pair of well sclerotized spiral structures (Fig 4A–4D) similar to the "spiral organs" in *Veigaia* [7, 30]. The left internal spiral structure has a clockwise direction and the right has a counterclockwise direction starting from the ventral pore to deep within the body which is easier to discern in the rota table **3D model** (Fig 4D). The total average length from the pore to the end of the sclerotized spiral part is 1285 microns (N = 9; 1165—1205).

Based on orthogonal views of the 3D images, the sperm reception system seems to be an invagination of the epicuticle of the coxae. Internally, these spiral structures connect to an elaborate internal secondary insemination system closely resembling homologous structures in other Dermanyssina. Each of the sclerotized structures continues into an unsclerotized structure resembling the sacculus vestibulus in the phytoseiid sperm induction type [2]. Then the sacculus vestibulus-like structure connects to a major duct, which connects to an atrium and a calyx (Fig 6A–6D arrows) and finally connects to an inconspicuous vesicle that occupies a large part of the abdomen. Additional minor duct(s) were not observed.

## 2.2 Matching and function

This study confirms that the female internal secondary genital structures match the external structures of the male in shape, direction, and size. The female internal spiral structure has the same direction and shape as the male spermatodactyl in the venter-to-venter mating posi-tion adopted by *M. colossus* (Fig 7A and 7B). In addition, the total length of the spermatodac-tyl is similar to the length of the sclerotized part of the female sperm induction system (averages of 1180 $\mu$m and 1115 $\mu$m, respectively). Notably, we do not have evidence that the male spermatodactyl is fully inserted in the female internal spiral structure, but given the

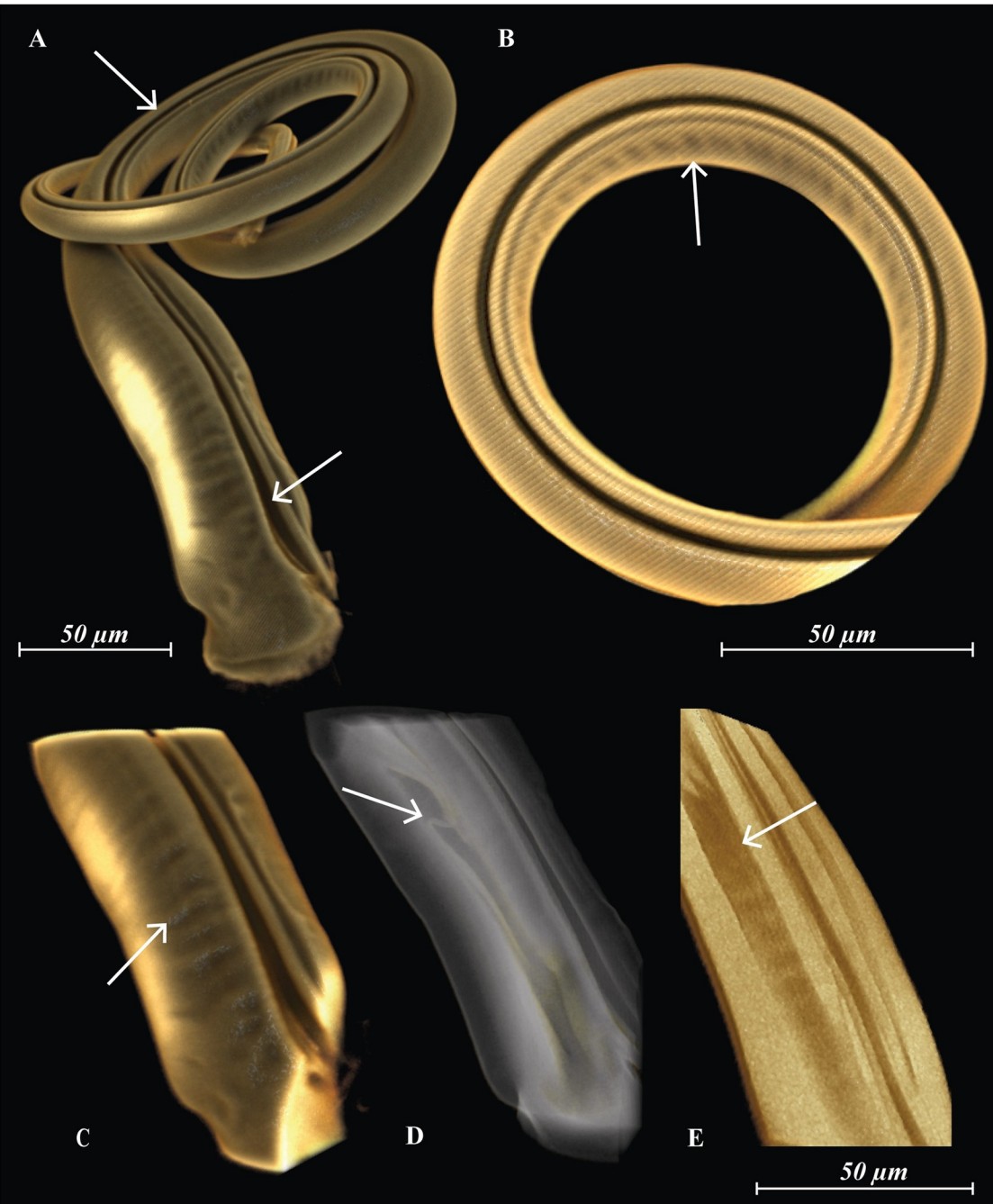

**Fig 3. *Megalolaelaps colossus* male spermatodactyl based on CLSM.** (a) external groove (arrows) (b) detail of the distal annuli-like structures (arrow) (c) detail of the proximal annuli-like structures (arrow) (d) detail of the unidentified structure unfolded (arrow) (e) detail of the unidentified structure folded (arrow).

complexity and matching direction, shape, and size, which is easier to percieve in the **3D models** (Models Fig 7A male and Fig 7D female), such insertion is expected. This near perfect match suggests the need for an explanation. The observations are consistent with a hypothesis of genital coevolution.

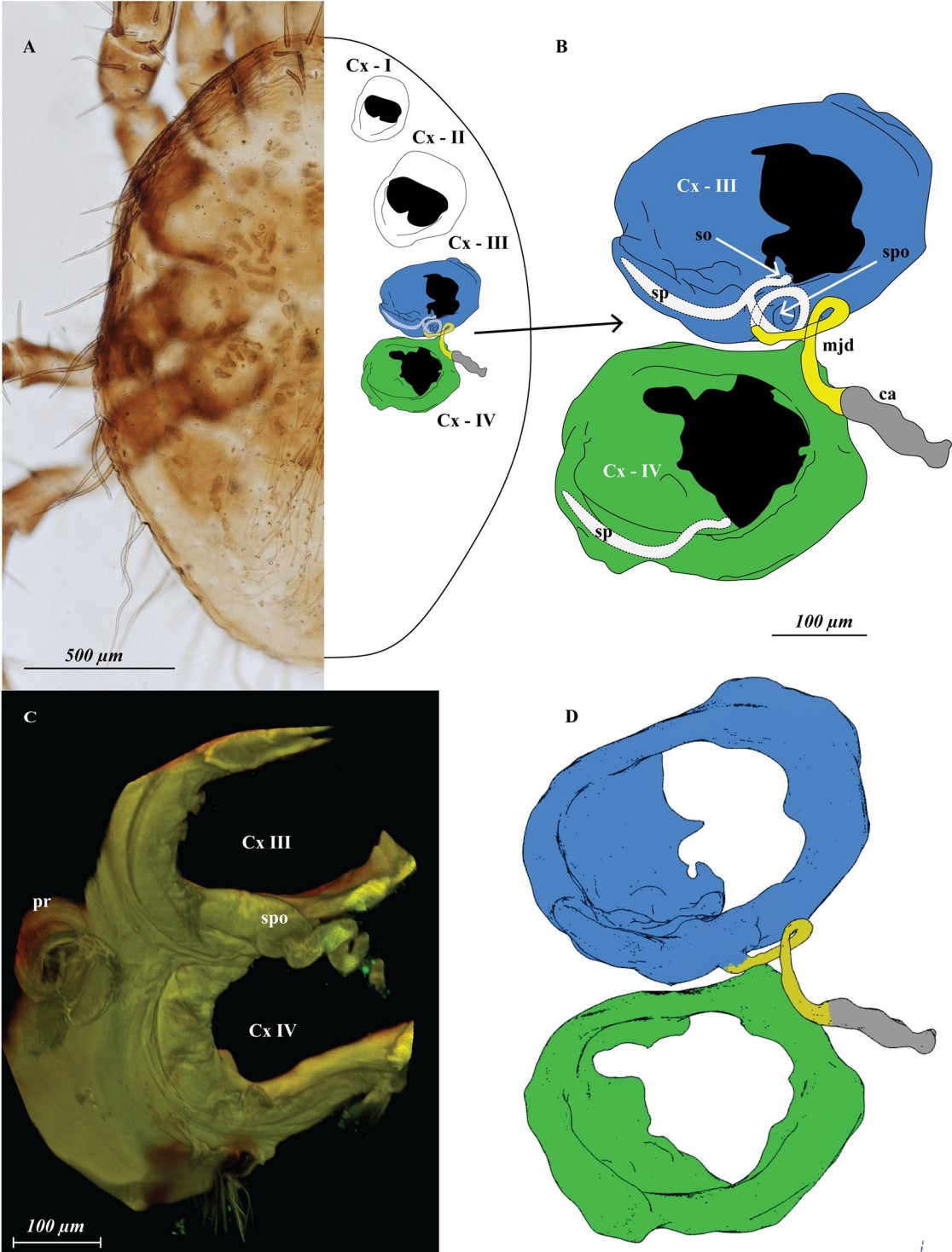

**Fig 4.** ***Megalolaelaps* females.** (A) schematic representation of *Megalolaelaps colossus* female's sperm access system (B) detail structure of coxa III and IV. (C) Slide prepared in 1930 with Hoyer's medium of *Megalolaelaps enceladus*. Region of coxae III–IV from dorsum to venter, rendered model in FIJI image using CLSM. (D) Screenshot of 3D rotatable model of *Megalolaelaps colossus* (S1 File) female's sperm access system with coxa III in blue and coxa IV in green. ca, calyx; Cx I—IV, coxa I—IV; mjd, major duct (yellow); pr, peritreme; so, solenostomes; sp, sperm pocket (dotted); spo, spiral organ.

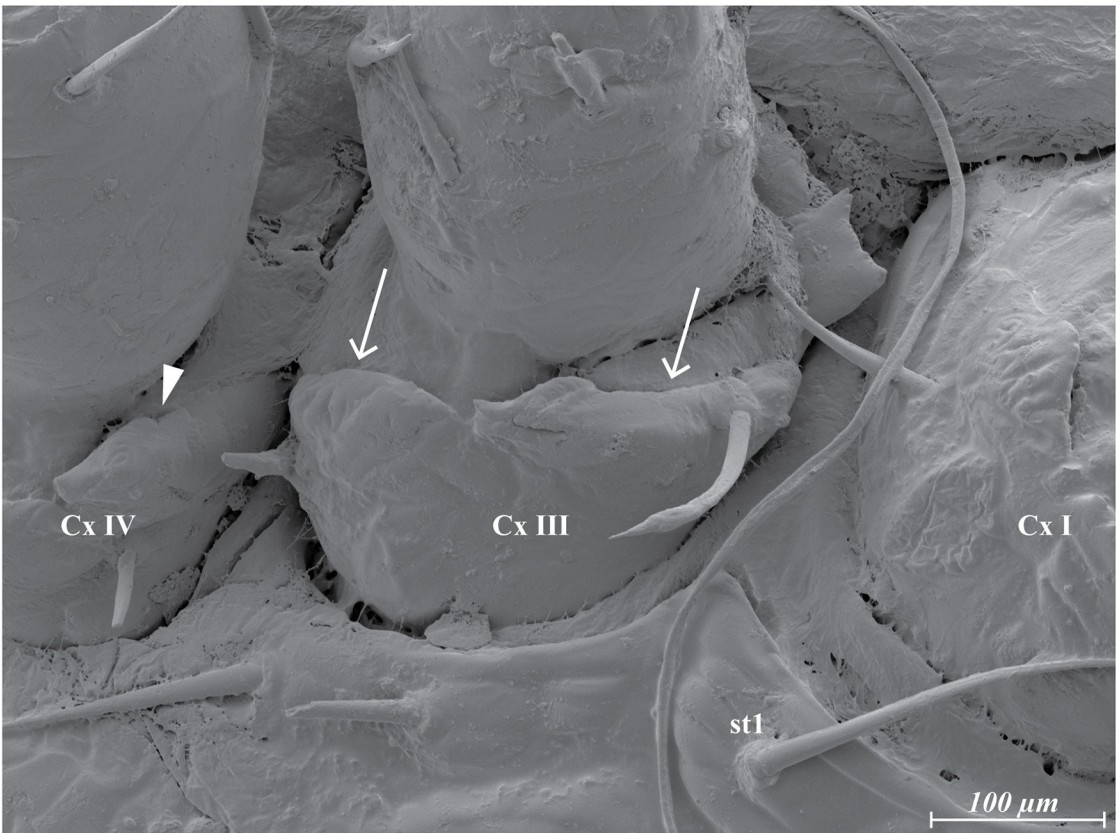

**Fig 5.** *Megalolaelaps colossus* **female ventral LT-SEM image of coxae III—IV.** Cx I—III, coxa I—III; st1, sternal setae 1. Possible solenostomes (arrows).

### 2.3 LT-SEM in Mesostigmata

LT-SEM requires minimal specimen preparation and enables specimens with high water content to be frozen in liquid nitrogen and placed in the vacuum of the SEM and observed in a near natural state without the use of chemical fixatives.Cryo-preparation of mites in combination with the use of high-resolution field emission SEMs enabled us to observe secondary genital pores located on coxae III (Fig 5 arrows) and pores of dead-end pockets or putative sperm reservoirs on coxae IV (Fig 5 arrowhead) of the females, and for males, the membrane covering the connection between the spermatodactyl and the movable digit (Fig 2D arrow). Light microscopy (DIC nor Phase Contrast) did not reveal these structures.

### 2.4 CLSM in Mesostigmata

The steps and variables to be considered when using CLSM for internal and external structures in Mesostigmata are as follows. Collecting methods can follow standard recommendations and protocols (e.g. [2]) because this will not affect the result of the final scan. However, preservation and clearing methods may have a significant impact. For instance, clearing specimens with lactic acid or KOH may cause degradation of internal organs and soft cuticle. The cuticle of mites tends to have a relatively strong level of autofluorescence compared to other arthropods, possibly because of elevated levels of components such as pteridines and resilin [18].

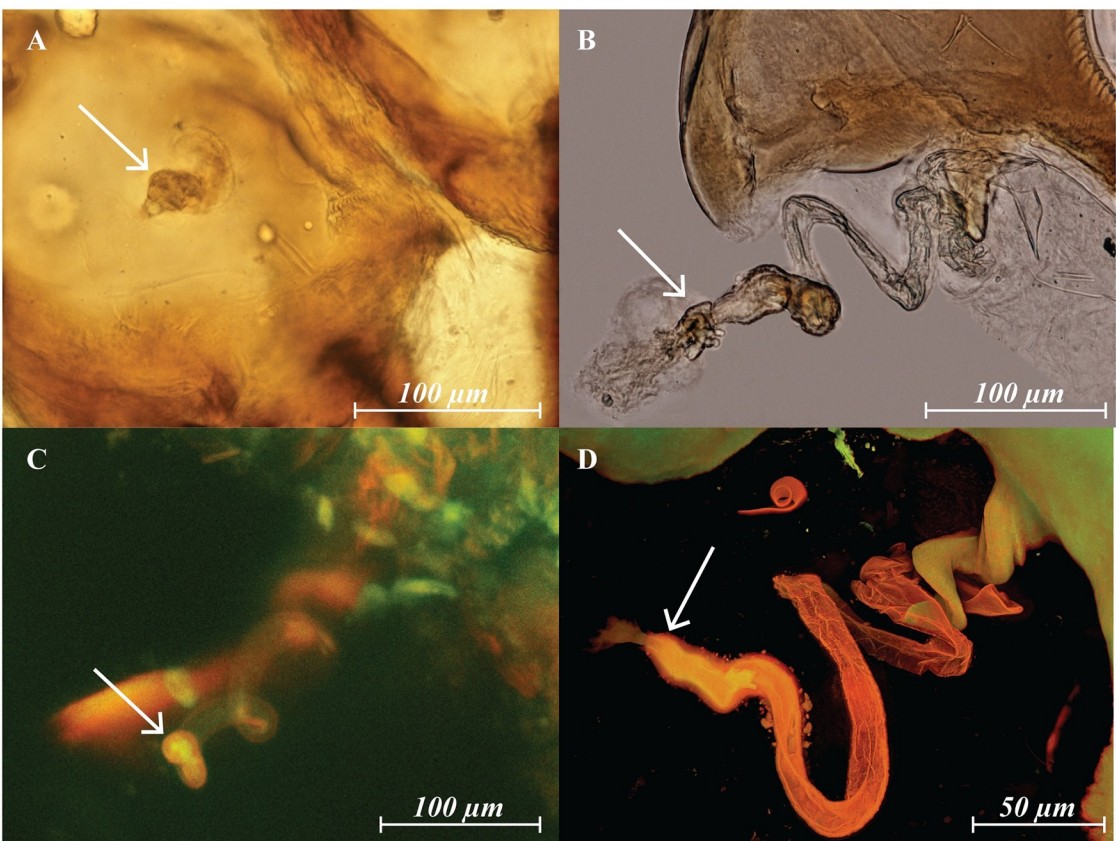

**Fig 6. *Megalolaelaps colossus* female secondary insemination system.** (A) DIC Image of cleared specimen, dorsal view (B) DIC image of a dissected specimen (C) MIP images using CLSM of cleared specimen (D) MIP image using CLSM of a dissected specimen.

Therefore, use of targeting fluorochromes is only necessary for unsclerotized structures such as muscle or nervous tissue. Autofluorescence signal strength tends to diminish over time and also with exposure to fluorescent light, an effect known as photo bleaching [18]. However, Mesostigmata preserved in a slide mounting medium such as Hoyer's seem to retain a good amount of autofluorescence and make good quality images. For instance, we photographed one female of *M. enceladus* Berlese using a slide prepared in 1930 with Hoyer's medium (Fig 4C). Additionally, photobleaching does not appear to be a significant problem with Mesostigmatid mites. Some of the specimens were scanned more than ten times, and the image quality did not degrade. Furthermore, DNA extraction in mites might enhance auto-fluorescence after treatment with proteinase K [27]. Before starting a CLSM session, a fluorescence stereomicroscopy image was acquired in order to check which filters are most suitable for each specimen (ventral view S1A–S1D Fig; dorsal view S2A and S2D Fig). However, these techniques continuously expose the mites to high intensities of UV light which could accelerate photo bleaching. There appears to be a correlation between the level of sclerotization and the optimal channel. For hard cuticle, red seems to be optimal (S1C, S2C and S3C Figs), for setae and soft cuticle blue seems to be better (S1A, S2A and S3A Figs), while the green channel appears to be more versatile than the other two channels because it highlights both soft and hard cuticle (S1B, S2B and S3B Figs). These observations complement the findings for well

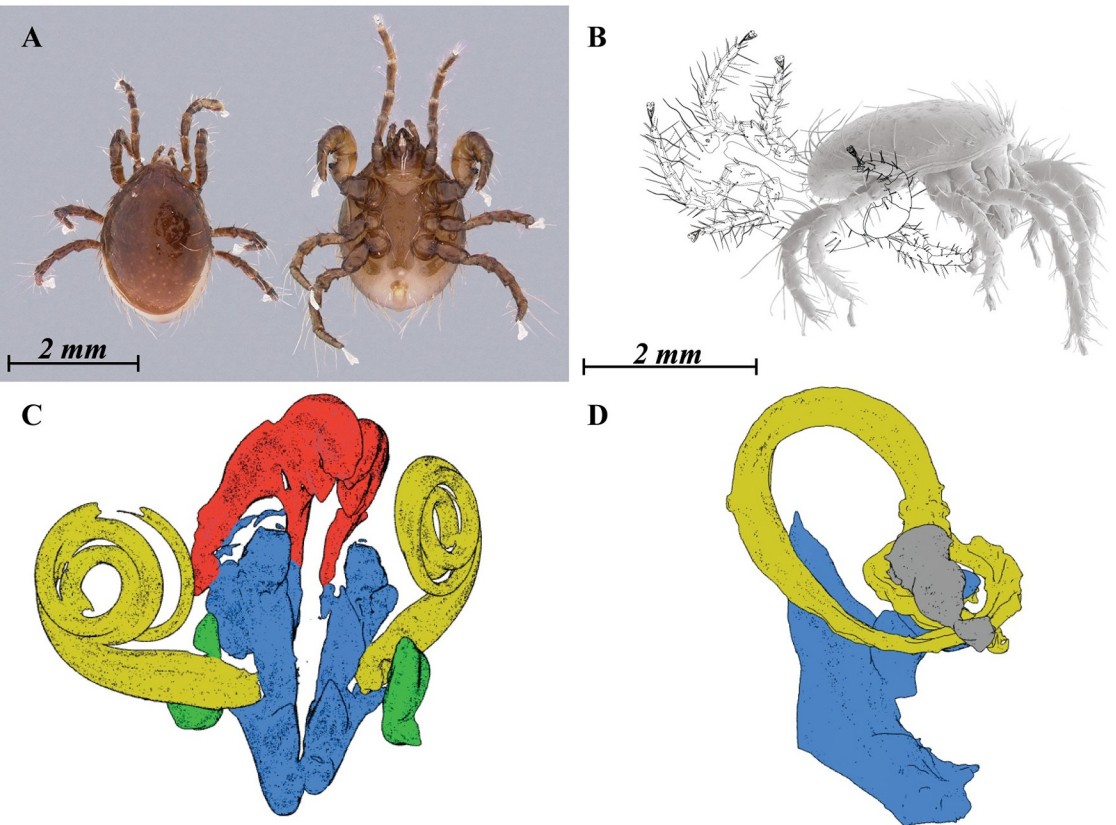

**Fig 7. Copula model for *Megalolaelaps colossus* (A) dorsal view female and ventral view male, stereomicroscope images (B) female and male in copula, lateral view (C) Screenshot of 3D rotatable model of male spermatodactyl (S2 File) (D) Screenshot of 3D rotatable model of female secondary insemination system (S3 File).**

sclerotized acariform mites (e.g., *Carabodes* [21], or *Torrenticola* [27]) and for other Mesostigmata (i.e. *Dermanyssus* mentioned in [31]. Overall, a combination of all three colors is recommended for any study in external and internal morphology in Mesostigmata (S1–S3 Figs).

## 3 Discussion

### 3.1 CLSM in Mesostigmata

Only two previous studies have used CLSM in Mesostigmata, one for the study of olfactory systems in the predatory mite *Phytoseiulus persimilis* [32] and another to study the diet of the honey bee mite *Varroa destructor* [33]. This is the first time CLSM is used for a comparative study of genital systems. Although specific parameters needed to be adjusted (see S1 Table), we demonstrate that CLSM is indeed a quite useful method to study internal and external anatomy in Mesostigmata. CLSM allowed 3-dimensional reconstruction of the female and male secondary genitalic structures. The female secondary genitalic structures in *M. colossus* are highly complex, including various pockets and a pair of large, sclerotized, spiral structures connecting with the secondary genital openings near coxae III. As such they match the male secondary genitalic structures in complexity. It is important to note that these female secondary genitalic structures are in addition to the "standard" unsclerotized ducts of the dermanyssine genitalia as described by [3, 6–9].

The mechanism leading to genital coevolution in this species is less clear. Although genital coevolution has been studied in many invertebrates [5, 14], the current system most closely resembles some of the genitalic matching systems in vertebrates [16, 17, 34]. For example, the long and coiled vaginal "elaborations" in ducks correlate with the morphology of the phallus. That system has been hypothesized as resulting from sexual selection, specifically sexual conflict [17]. The authors of that study [17] came to that conclusion after contemplating several alternative explanations for the observed matching pattern of genitalia, specifically natural selection, genitalic homology, and several sexual selection mechanisms such as male competition, female choice and sexual conflict [17]. Below we consider the fit to our data of these, and other, alternatives for *Megalolaelaps*. First, **natural selection** could explain the development of complex male and female genitalia if the relevant traits were subject to independent selective pressures that could lead to similar morphologies. For example, the spermatodactyl of the male could have evolved as a feeding accessory (unlikely given that the big coiled spermatodactyl may actually interfere with feeding). Similarly, the long internal spirals of the secondary insemination system of the female could have evolved as a mechanism to avoid contamination during copula since these mites live and reproduce in manure. However, this scenario alone does not explain why male and female structures mirror each other in shape, direction, and size.

Second, **homology** could explain the extreme and matching morphologies if the affected organs in both sexes were indeed homologous. However, while the primary reproductive elements (e.g. testis, ovary) are homologous for males and females, the spermatodactyl and sperm induction system have different locations and origins (procuticle of the chelicera and epicuticle of the coxae, respectively).

Among various mechanisms related to sexual selection, a lock-and-key model would seem to be an excellent explanation for the observed matching. The idea is that close matching of female and male genitalia would evolve to avoid inefficient matings with heterospecific partners. However, lock-and-key does have some strong predictions, including an expectation of reproductive character displacement in areas of sympatry of multiple closely related species. Available data allow a preliminary test of this prediction. Most of our observations are based on specimens of *M. colossus* from Quindio, where only *M. colossus* occurs. However, we (OCH, HK) also collected *M. colossus* in the Amazon region, where it co-occurs with a new species *M.* n. sp. 2. Male *M. colossus* at that side have a longer spermatodactyl than *M.* n. sp. 2. (960 $\mu$m vs. 773 $\mu$m) but these spermatodactyl are smaller than those of *M. colossus* males from Quindio, the reverse of what would be predicted under the lock-and-key model for a region of species overlap. This leaves as the least unlikely hypothesis sexual conflict [35]. Sexual conflict has been invoked as a major force driving the antagonistic coevolution of physiological, behavioral and/or morphological traits for reproduction [36] and assumes selection in both partners for features that may lead to reproductive dominance over the other, leading to an evolutionary arms race. Under this model if a male has a complex reproductive morphology, female genitalia are assumed to coevolve in complexity to recover control over reproduction. Practically, that would imply that female reproductive structures should show equivalent complexity to that shown in their male counterpart (e.g. size, shape, direction, motion, location among others). These predictions appear to match our observations of morphology and biology of *M. colossus*. Of course this hypothesis brings up more questions. For example, is there any conflict between male and female in this species, and if there is, how that would be expressed? Rearing of the mites [37] has solved multiple questions, but not (yet) that one. A final note on the possibility of sexual conflict in this system concerns the presence of a pair of secondary pores on coxae IV. These pores have a similar configuration as the pores on coxae III, but they lead to dead-end sperm pocket-like structures and well sclerotized (but smaller) spirals (Fig 4B). These structures might result from serial duplication/homology without any direct

significance, but they would also fit into a sexual conflict model as deflection options for females trying to avoid mating with non-preferred males. Our observations raise a second set of questions, dealing with the mechanics of mating in *M. colossus*. While the match of male and female genital structures suggests extensive introduction of the spermatodactyl into the female secondary insemination system, how is this done given the strong sclerotization of both male and female systems? In fact, there are two, somewhat, separate questions: 1) How can the tightly wound structure of the male spermatodactyl be introduced in the spiral of the female structures, and 2) what can be the mechanism to power that introduction? The answer to the first question is relatively straightforward. The articulation of the spermatodactyl with the movable digit, as well as the ring- or annulus-like structures observed in the spermatodactyl proper should allow considerable flexibility of the spermatodactyl during insertion into the long coiled female secondary insemination system. Laboratory experiments with the spermatodactyl demonstrated that the articulation is movable and also that the spermatodactyl can indeed unfold from its coiled resting state (Fig 8). The same experiment (S1 Video) showed that the spermatodactyl resumed its coiled shape after release, suggesting that coiling results from elasticity. As for the second question, this is much less clear. Insertion by introducing the tip of the spermatodactyl and continuous pressure through the male chelicera seems unlikely, as it is unclear how such pressure could uncoil the spermatodactyl. That would suggest some pressure from within the male chelicera to effect uncoiling. One option would be musculature in, or at the base of, the spermatodactyl. Potential muscle fiber-like structures were observed in *M. enceladus* under CLSM (S5 arrow) but presence of such fibers could not be confirmed for *M. colossus*. It is also unclear what the origin of such muscles, if present, would be. An alternative would be hydrostatic pressure, perhaps created in the movable or fixed digit and expressed through the cavities in the spermatodactyl. Unfortunately, available evidence is insufficient to test such a hypothesis. In summary, *Megalolaelaps colossus* has highly complex secondary genital structures in both sexes, with structures in females and males that are largely matching. To resolve the mechanism for this case of apparent genitalic coevolution will require substantial additional data, but with the advent of new microscopic techniques we have the ability to assemble the basic data to start asking such questions.

## 4 Materials and methods

### 4.1 Collecting data

Mites were found associated with dung beetles, *Oxysternon conspicillatum* (Weber) in two localities. A mixed crop of coffee and plantains located in "El Bosque" farm, Calarcá, Quindío, Colombia, 4˚31'09.7"N, 75˚37'35.9"W, and Esteban Carillo's farm, Km 11 Tarapacá road, Leticia, Amazonas, Colombia (4˚05'44.7" S 69˚57'00.9" W). Beetles were collected using non-lethal pitfall traps. Mites were removed under a dissecting scope and preserved in 95 percent ethanol. Specimens studied in this work have collecting permits with resolution number 374 of March 7, 2014, Art. 1. issued by the CRQ (Corporación Regional Del Quindío), which grants framework permission for study collection scientific research purposes with non-commercial biological diversity.

### 4.2 Light stereomicroscopy (St)

Images were obtained using a Leica Z16 APOA stereomicroscope (Buffalo Grove, IL) with a 1X objective lens and equipped with a JVC KY-F75U digital camera (Hachioji, Japan). Stacked images were merged and processed using Combine ZP (https://combinezp.software.informer.com/) and AutoMontage Pro (http://www.syncroscopy.com/) at the C.A. Triplehorn Insect Collection (OSUC) of the Ohio State University (OSU).

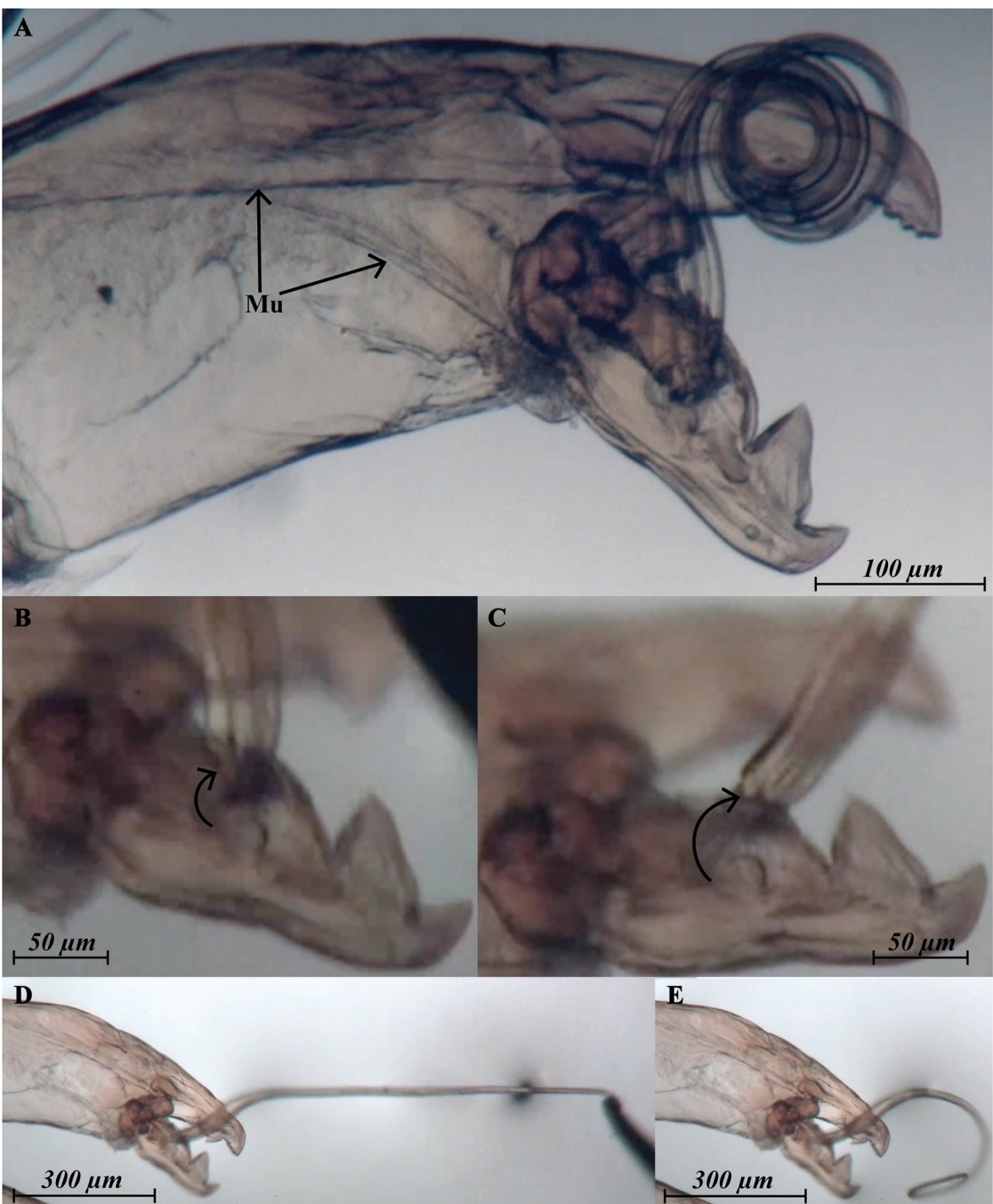

**Fig 8.** *Megalolaelaps colossus* **male spermatodactyl under dissecting scope (A) detail of muscles (B) articulation retracted (C) articulation extended (D) spermatodactyl stretched (E) spermatodactyl going back to normal position.** Mu, muscle.

### 4.3 Fluorescence stereomicroscopy (FSt)

Images of the mites were obtained by placing them in a petri-dish with 70 percent ethanol. A Zeiss AxioZoom microscopy system (Thornwood, NY) at the Electron and Confocal Microscopy Unit (ECMU) at the USDA-ARS in Beltsville, Maryland. The images were obtained using

a 1x 0.25NA PlanNeoFluor objective. Fluorescence microscopy was accomplished using a 200 watt mercury vapor lamp (HXP Short Arc Lamp, Thornwood, NY) with a filter set for DAPI with excitation at 335–383nm, beam splitter 395nm and emission at 420–470m; GFP with excitation at 450–490nm, beam splitter 495nm and emission at 500–550nm; mRFP with excitation at 559–585nm, beam splitter 590nm and emission at 600–690nm. Fluorescence was captured using an AxioCam 506 mono camera. Zen 2 Pro Blue (Thornwood, NY) 64-bit software was used to capture 15–20 Z-stack images using extended depth of focus to produce 2D images.

## 4.4 Light microscopy

Mites were cleared with lactic acid, dissected and mounted in Hoyer's medium on microscope slides and observed using differential interference contrast microscopy (DIC) using a Nikon Eclipse 90i (Melville, NY) microscope equipped with a motorized head with 10X, Plan Apo 20X, 40X, 40Xoil, and 100Xoil objectives and a PC controlled Ds-5M-U1 digital camera at the Acarology Laboratory of the Ohio State University (OSAL).

## 4.5 Scanning electron microscopy (SEM)

Mites were directly placed on aluminum studs on a disk-sample holder of a Hitachi TM3030Plus Tabletop SEM (Tarrytown, NY) equipped with a proprietary, highly sensitive low-vacuum secondary electron detector. The low-vacuum mode allowed us to image the mites without any coating or dehydration processing.

## 4.6 Low temperature scanning electron microscopy (LT-SEM) (ECMU)

Mites were observed through LT-SEM as described in Bolton et al. [38]. Briefly, mites preserved in ethanol were allowed to dry for a short time then secured to 15 mm x 30 mm copper plates using ultra smooth, round (12 mm diameter), carbon adhesive tabs (Electron Microscopy Sciences, Inc., Hatfield, PA, USA). The specimens were frozen conductively, in a Styrofoam box, by placing the plates on the surface of a pre-cooled (-196˚C) brass bar whose lower half was submerged in liquid nitrogen (LN2). After 20–30 s, the holders containing the frozen samples were transferred to a Quorum PP2000 cryo-prep chamber (Quorum Technologies, East Sussex, UK) attached to an S-4700 field emission scanning electron microscope (Hitachi High Technologies America, Inc., Dallas, TX, USA). The specimens were etched inside the cryo-transfer system to remove any surface contamination (condensed water vapor) by raising the temperature of the stage to -90˚C for 10–15 min. Following etching, the temperature inside the chamber was lowered below -130˚C, and the specimens were coated with a 10nm layer of platinum using a magnetron sputter head equipped with a platinum target. The specimens were transferred to a pre-cooled (-130˚C) cryo-stage in the SEM for observation. An accelerating voltage of 5kV was used to view the specimens. Images were captured using a 4pi Analysis System (Durham, NC).

## 4.7 CLSM

Images were obtained using three different confocal laser scanning microscopes systems, with mites mounted in different conditions as follows: at the Campus Microscopy and Imaging Facility (CMIF) at OSU we used an inverted Zeiss LSM880 microscope with Airyscan and one Olympus FV1000-Filter confocal system equipped with an upright Olympus BX61F/BX62; at the ECMU images were obtained using an inverted Zeiss LSM710 system (Thornwood, NY) mites were on microscope slides with Hoyer's medium, or in a concavity slide with glycerin, or

placed between two coverslips with glycerin as described in Gulbronson et al. [39]. The LSM710 used 3 lasers simultaneously: a 405nm Diode laser, 488nm Argon laser, and a 561 DPSS laser, with a pin hole of 30 microns, and a filter set capturing emission between 410–483nm for blue; 495–553nm for green and 566–703nm for red to obtain 82–354 Z-stack images. The Olympus FV1000-Filter system used 4 lasers simultaneously: a 405nm Diode laser, 488nm Argon laser, a 543nm HeNe1, and a 633nm HeNe2, with a pin hole of 120 microns and a filter set capturing emission between 430–470nm for blue, 505–525nm for green, 560–660nm for yellow-orange, and 655–755nm for red to obtain 71–181 Z-stack images. The LSM880 used 1 or 2 lasers simultaneously: a 405nm Diode laser, and a 561 DPSS laser with a pin hole of 35 microns passing through a MBS 488/561/633 main dichroic beam splitter filter and a MBS 405 filter, with detection filters 459 and 578 Airyscan to obtain 483–503 Z-stack images. Complete parameters for individual images are included in S1 Table with their respective Morph Bank ID [40].

### 4.8 Image enhancement

LT-SEM grey scale images have allow scientist to observe ultra-delicate external structures that have not been previously observable [38]. Black and white SEM images can be colorized to resemble the natural color of the organism or colors can be added to emphasize critical morphological features [33]. However, although the LT-SEM allows for detailed studies of the external structures of mites it is not useful for studies of internal structures.

Z-stack images in Zeiss (.czi) or Olympus format (.oib) were opened and processed with the free software FIJI (Version 2.0.0; [41]), using the free-hand tool and the plugin 3D Scrip interactive animation [42] to isolate the region of interest (ROI), then the stack was saved in .tiff format. Tiff images were processed in two ways: 1) to convert Z-stack images into 3D models, FIJI was used to make the image binary, built the surface model and exported it in .stl format; then the .stl file was processed in Mesh Lab version 2020.06 [43] to remove unwanted regions, smooth surfaces, and to export the model to .u3d format to make it compatible with pdf. 2) using the open-source scientific visualization software Drishti [44]. In Drishti ImportTM, the tiff image was opened, and the histogram adjusted, after which it was saved in Drishti format (.nc). This image was opened in DrishtiTM for volume rendering. In this interface, opacity, density and gradient could be modified using the transfer and opacity tools to depict the ROI in high resolution. Then, when the rendered visualization was optimal, it was saved and exported as an .png image. Two dimensional images such as Maximum Intensity Projection (MIP) from stack, SEM, or LT-SEM or .pgn format from DrishtiTM, were modified in Photoshop Adobe Photoshop (Version 21.1.1; Adobe Systems Inc., San Jose, CA).

## Supporting information

**S1 Fig.** *Megalolaelaps colossus* **female ventral view.** Fluorescence stereomicroscopy image: (A) blue filter (B) green filter (C) red filter (D) combined.
(TIF)

**S2 Fig.** *Megalolaelaps colossus* **male dorsal view.** Fluorescence stereomicroscopy image: (A) blue filter (B) green filter (C) red filter (D) combined.
(TIF)

**S3 Fig.** *Megalolaelaps colossus* **larvae.** MIP images of tarsus I with CLSM: (A) blue channel (B) green channel (C) red channel (D) combined.
(TIF)

**S4 Fig.** *Megalolaelaps colossus* **male.** Leg II: (A) colorized LT-SEM of lateral view (B) drawing the detail of leg II (C) colorized LT-SEM image.
(TIF)

**S5 Fig.** *Megalolaelaps enceladus* **male.** Spermatodactyl and muscle fibers (arrow). Co, corniculi; fd, fixed digit; md, movable digit; Mu, muscle; pdp, pedipalp; Sd, spermatodactyl.
(TIF)

**S1 Video.** *Megalolaelaps colossus* **male.** video: spermatodactyl under dissecting scope with detail of muscles, articulation and its motility. Demonstrates how the spermatodactyl can be "unfolded" and how it returns back to normal position by elasticity.
(MOV)

**S1 File.** *Megalolaelaps colossus* **female.** 3D rotatable model of female's sperm access system with coxa III in blue and coxa IV in green.
(PDF)

**S2 File.** *Megalolaelaps colossus* **male.** 3D rotatable model of male's chelicerae with spermatodactyl in yellow, tip of fix digit in red, movable digit in blue and corniculi in green.
(PDF)

**S3 File.** *Megalolaelaps colossus* **female.** 3D rotatable model of female's secondary insemination system with coxa III in blue, major duct in yellow, and calyx in grey.
(PDF)

**S1 Table. Individual parameters for the images.** All figures and 3D models are available at
http://morphobank.org/permalink/?P3717.
(PDF)

## Acknowledgments

Thanks to Konstantin Thierbach and Genevieve Penzone for their valuable help with image processing. Thanks to Andrew Ulsamer, Debra Creel and Joseph Mowery for helping with specimen preparation, imaging, literature, and loan of specimens.

## Author Contributions

**Conceptualization:** Orlando Cómbita-Heredia, Edwin Javier Quintero-Gutiérrez, Hans Klompen.

**Data curation:** Orlando Cómbita-Heredia.

**Formal analysis:** Orlando Cómbita-Heredia.

**Funding acquisition:** Orlando Cómbita-Heredia, Hans Klompen.

**Investigation:** Orlando Cómbita-Heredia, Ronald Ochoa, Edwin Javier Quintero-Gutiérrez, Gary Bauchan, Hans Klompen.

**Methodology:** Orlando Cómbita-Heredia, Connor J. Gulbronson, Ronald Ochoa, Gary Bauchan.

**Project administration:** Orlando Cómbita-Heredia.

**Resources:** Orlando Cómbita-Heredia.

**Software:** Orlando Cómbita-Heredia.

**Supervision:** Orlando Cómbita-Heredia, Hans Klompen.

**Validation:** Orlando Cómbita-Heredia, Hans Klompen.

**Visualization:** Orlando Cómbita-Heredia, Hans Klompen.

**Writing – original draft:** Orlando Cómbita-Heredia.

**Writing – review & editing:** Orlando Cómbita-Heredia, Ronald Ochoa, Gary Bauchan, Hans Klompen.

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
