## [Decision Letter · Decision Letter 0]

25 Nov 2020

PONE-D-20-28641

Size, shape, and direction matters: recognizing genital coevolution using multi-microscopy technologies and 3D modeling

PLOS ONE

Dear Dr. Combita-Heredia,

Thank you for submitting your manuscript to PLOS ONE. After careful consideration, we feel that it has merit but does not fully meet PLOS ONE’s publication criteria as it currently stands. Therefore, we invite you to submit a revised version of the manuscript that addresses the points raised during the review process.

I agree with both reviewers that your study brings forth novel data and interpretations of the genital morphology of a mesostigmatid mite using a broad range of microscopic techniques. Both reviewers, however, raise a number of reservations that relate to the interpretations of your results, as well as the presentation of the paper. These important concerns need all be carefully heeded in a rewritten manuscript. Please pay particular attention to the potentially misleading title, the focus on evolutionary interpretations that may not be fully warranted given this one species system, and the recommendations of the reviewers how to rewrite parts of the text, and relabel the figures.

We look forward to receiving your revised manuscript.

Kind regards,

Matjaž Kuntner

Academic Editor

PLOS ONE

Journal Requirements:

2. In your Methods section, please provide additional information regarding the permits you obtained for the work. Please ensure you have included the full name of the authority that approved the collection sites access and, if no permits were required, a brief statement explaining why.

'..This project was partially funded by the fellowship `Doctorados Exterior' 568-2012- C.C. 80048516 Colciencias, Colombia (OCH). Mention of trade names or commercial products in this publication is solely for the purpose of providing specific information and does not imply recommendation or endorsement by the U.S. Department of Agriculture. USDA is an equal opportunity provider and employer. Some images presented in this report were generated using the instruments and services at the Campus Microscopy and Imaging Facility. This facility is supported in part by grant P30 CA016058, National Cancer Institute, Bethesda, MD.'

'(OCH) Doctorados en el Exterior Grant 568-2012- C.C. 80048516 Colciencias, Colombia. https://minciencias.gov.co/quienes_somos/sobre-colciencias '

5. Please ensure that you refer to Figure 8 in your text as, if accepted, production will need this reference to link the reader to the figure.

6. Please include a copy of Table 6 which you refer to in your text on pages 6 and 16.

Reviewers' comments:

Reviewer's Responses to Questions

**Comments to the Author**

1. Is the manuscript technically sound, and do the data support the conclusions?

Reviewer #1: Partly

Reviewer #2: Yes

2. Has the statistical analysis been performed appropriately and rigorously? 

Reviewer #1: N/A

Reviewer #2: N/A

3. Have the authors made all data underlying the findings in their manuscript fully available?

Reviewer #1: Yes

Reviewer #2: Yes

4. Is the manuscript presented in an intelligible fashion and written in standard English?

Reviewer #1: Yes

Reviewer #2: Yes

5. Review Comments to the Author

Reviewer #1: The reviewed MS is focused on the investigation of secondary genital structures in a mite species Megalolaelaps colossus (Mesostigmata), with the aid of conventional light microscopy and modern microscopy techniques (LTSEM and CLSM). The main goals of the paper are “…to use multiple microscopy technologies to investigate the following hypotheses: 1) the female secondary genital structures are as complex as the male structures, and 2) the secondary insemination system in females of M. colossus matches the complexity of the male spermatodactyl, as expected under a hypothesis of genital coevolution”.

The authors discovered a topographical resemblance between male (spematodactyl, SD) and female (secondary genital channel) secondary genital structures. Based on this observation, the authors concluded “…the female internal genital structures match the male external structures, in accordance with expectations under a genital coevolution scenario.” Probably, it was this conclusion that was the reason for the very bright title of the reviewed MS - “Size, shape, and direction matters: recognizing genital coevolution using multimicroscopy technologies and 3D modeling.”

Note, that it is slightly unexpected that “secondary genital structures” are not mentioned in the title, and that Abstract is focused on the “genital coevolution model” and “the mechanics of mating and sexual selection”. These “code words” (or “key terms”) seems slightly conflicting with the morphological and methodological content dominating this MS. Perhaps testing the genital hypotheses (mentioned in the introduction of the MS) is somewhat artificial. Instead, it might be more appropriate to focus on the morphology and methodology (the strongest aspect of this MS): the protocol for CLSM, the characteristics of the cuticle of mesostimatan mites, and the topography of the genitalia in the model species. In its current format, the MS is vague and, in my opinion, not focused, with the main interesting findings hidden under reasoning about general “genital concepts”. Additionally, the title of the MS is too broad and seems reflecting the results not entirely accurate.

In general, data on the structure of secondary reproductive structures (especially in the female) could be given more clearly. For a better understanding, the reader definitely needs a scheme showing all structures.

I have some doubts about the main results of the study. As it follows from the text, the authors state that the long coiled spermatodactyl (SD) matches the long spiral secondary genital (SG) channel of females and assume that this is because of the coevolution of male and female SG structures. However:

1) This idea is based on the hypothesis that spematodactyl is ENTIRELY inserted into the SG channel. As far as I know, the complete insertion of SD into female SG channel has never been proved/observed in mesostigs, suggesting the possibility that only the tip of SD could be used for sperm transfer.

2) Long SD could be important not only for complete insertion but also for accommodating differently sized males and females, because there is always a variation in a population so that both smaller and bigger males/females are present. Therefore, the SD should be considerably longer than the minimal distance to the external pore, otherwise, some males will not be able for sperm transfer.

3) The spiral shape could have independently evolved in both sexes because of parsimony reasons. It is much easier to live with a compact spiral structure than with a long linear SD. Additionally, a spiral form is a universal way to make a structure mechanically stronger, which is especially important for insert structures and different internal tubular structures.

4) The sperm pocket is situated very close to the external pore and it is not clear why the SD should be inserted entirely.

5) When the SD is being inserted it should change its coiled shape, therefore any real “matching” between male and female spiral shapes disappear.

The discovery that CLSM is a powerful tool for studying external and internal structures of mesostigs is a very strong point of the reviewed MS. Although this aspect is described quite well, some broader discussion, including comparison with results obtained by previous authors, would make the MS stronger. Perhaps, the authors could compare their results with papers by Valdecasas, Cramerik, Haug, Bolton, Chetverikov. Additionally, there is data in the literature (https://doi.org/10.11646/zootaxa.4066.3.4), suggesting that the green laser is the best for observing Dermanyssus (Mesostigmata) under CLSM. Remarkably, in this study the authors found the red laser to be more appropriate for mesostigs. This fact probably deserves a brief discussion.

I feel the structure of some parts of the MS needs revisions. Some paragraphs could be transferred from the Results / Discussion to the Introduction (e.g. general data on the usefulness of LTSEM/CLSM – p.10, lines 56-65, and “initial assumption” on CLSM based on the data from literature – p. 12, lines 139-141).

Besides, some new unexpected data/materials on the mites from Amazonia are mentioned in the Discussion (p. 17, lines 196-201) impeding a clear understanding of the text.

The figure captions, figure numbers, and arrows in the figures need careful check. There are also several repetitions (e.g. the phrase “The left internal spiral structure has a clockwise direction and the right has a counterclockwise direction” can be found several times in the text).

Maybe the authors could find the following paper interesting and relevant to the context of their study: http://dx.doi.org/10.1080/01647954.2013.813583

In general, I think that the MS needs careful reconsideration, partial rewriting, and restructuring and suggest “minor revision”.

Philipp E. Chetverikov

Saint-Petersburg

02/11/2020

Reviewer #2: The manuscript by Cómbita-Heredia et al describes the genital morphology of a mesostigmatid mite using different microscopic techniques. By doing so, the authors also compared different techniques focussing especially on the suitability of CLSM for the study of cuticular structures. I enjoyed reading the manuscript and also think it will add value to the acarological research community.

The manuscript is well written, but I feel that it would benefit from restructuring to better separate the methodological part from the part about genitalic morphology and evolution. This is especially true for the figures. For example, on p.5 the authors describe the different results using different mounting media. They refer to supplementary figures and also the very useful table. I suggest to present the results of their methodological survey in a single comparative plate (besides the supplementary material) that readers can see the pro and cons of the different approaches immediately. I have one more comment to the section about the LT-SEM in the results section. The authors claim that color can be added to emphasize critical features. I completely agree with it, but was wondering why the authors didn´t do it. Instead of highlighting critical features the authors applied color to give the impression of a natural look of the structures. For example, the application of color in Fig 5 is completely misleading and does not add any value but rather confuses here. Especially the dark brown makes it hard to see any details and thus I strongly suggest to remove the color in Fig 5! In general, I would suggest to highlight only structures of interest (one of many examples: Labarque et al 2017, Zool J Linn Soc 181:308-341).

As mentioned above, I strongly suggest to combine figures. For example, Fig. 1-3 can be certainly merged as each of them shows similar aspects. In this way the reader focusses only on one comparative figure showing the morphology of the spermatodactyl. Alternatively, the authors could consider to use Fig. 1 rather for a plate adressing the CLSM methodology and only merge Fig 2-3. Moreover, I was surprised about Fig. 7. I would have expected to see the interactive models of the spermatodactyl and secondary insemination system together with the respective figures, as e.g. 3D spermatodactyl combined with Figs. 2 and 3. Finally, the authors should also consider to move Figure 9 to the other figures of the spermatodactyl and combine it with Figure 4.

Regarding the 3D models I would strongly suggest to invest some more time into them. Why the authors did not use the chance to segment the different parts to provide a fully interactive 3D pdf model (one of many examples: Bicknell et al 2018, PLOS one, https://doi.org/10.1371/journal.pone.0191400, Fig. S1). Since the authors focus also on the methodology and since it is the first time applying CLSM/3D methods to mesostigmatid mites, the author could set some standards and provide a workflow which can be applied in future studies using the same methodology. In doing so, the authors could explore also semi-automatic segmentation tools like Biomedisa (https://biomedisa.de/;
https://www.nature.com/articles/s41467-020-19303-w), which can enhance such workflow.

I find the title misleading. You did not test whether size, shape and directions matter. Moreover, using a single species you can hypothese a genital coevolution, but I would be more cautious and suggest to change the title.

More minor comments:

- p.6, first paragraph Discussion, last sentence: What does "standard" mean? Please explain and explicitely describe what you mean here!

- p.8, line 164ff: It reads that the system described here cannot be compared with similar systems in invertebrates but rather with vertebrates. I would rewrite this part as there are numerous example of invertebrates considering sexual conflict and genitalic evolution (one example: Kuntner et al. 2009, Evolution 63:1451-1463).

- p.11, line 194ff: Please provide a figure showing the intra- and interspecific size difference fo the spermatodactyl! Did you correct the size of the spermatodactyl against body size?

- p.12, line 232: The figure number is missing.

- I was wondering whether the authors consider investigating a mating pair in copula by using MicroCT? In this way you can visualize the interaction of male and female genitalia during coupling in a direct way. I guess it would be worth adressing as interested readers might wonder about it.

- Please label the different structures in Fig. 6 and explain in the caption where the arrows point to.

6. PLOS authors have the option to publish the peer review history of their article (what does this mean?). If published, this will include your full peer review and any attached files.

Reviewer #1: **Yes: **Dr Philipp E. Chetverikov

Reviewer #2: No

---

## [Author Response · Author response to Decision Letter 0]

15 Jun 2021

Note, that it is slightly unexpected that “secondary genital structures” are not mentioned in the title, and that Abstract is focused on the “genital coevolution model” and “the mechanics of mating and sexual selection”. These “code words” (or “key terms”) seems slightly conflicting with the morphological and methodological content dominating this MS ADJUSTED . Perhaps testing the genital hypotheses (mentioned in the introduction of the MS) is somewhat artificial. Instead, it might be more appropriate to focus on the morphology and methodology (the strongest aspect of this MS): the protocol for CLSM, the characteristics of the cuticle of mesostimatan mites, and the topography of the genitalia in the model species DONE. In its current format, the MS is vague and, in my opinion, not focused, with the main interesting findings hidden under reasoning about general “genital concepts”. Additionally, the title of the MS is too broad and seems reflecting the results not entirely accurate.

The reviewer is correct in stating that the main data in the ms refer to morphology and methodology, but there was a reason why we used those technologies to investigate the secondary genital structures. This type of extreme morphology immediately brings up the question of how such structures could evolve. Preliminary examinations showed what looked like a match between external male genitalic structures and internal female ones, further focusing that question. Our study proceeded with investigations on whether the match was indeed as good as we thought, and on methods to visualize this match (the bulk of this paper), but that was done against the background of an evolutionary question. We cannot envision that natural selection would result in structures like the observed spermatodactyl, which left some form of sexual selection as the least unlikely explanation, with sexual conflict providing the best match. We have re-written the relevant sections to increase clarity and focus on the “meat” of the paper, the analysis of morphological structure, but we want to keep the evolutionary context 

In general, data on the structure of secondary reproductive structures (especially in the female) could be given more clearly. For a better understanding, the reader definitely needs a scheme showing all structures. DONE?

I have some doubts about the main results of the study. As it follows from the text, the authors state that the long coiled spermatodactyl (SD) matches the long spiral secondary genital (SG) channel of females and assume that this is because of the coevolution of male and female SG structures. However:

1) This idea is based on the hypothesis that spematodactyl is ENTIRELY inserted into the SG channel. As far as I know, the complete insertion of SD into female SG channel has never been proved/observed in mesostigs, suggesting the possibility that only the tip of SD could be used for sperm transfer.

Outside of this project we have evidence of insertion from mites in the Heterozerconidae. We have an image showing a broken off spermatodactyl in the secondary genital opening of a female. If the spermatodactyl is never inserted, it also makes no sense why it should be able to “unfold”, as we have demonstrated it can

More general, the reviewers alternative explanation does not explain why these structures in both male and female are so complex, or why they should be matching in structure

2) Long SD could be important not only for complete insertion but also for accommodating differently sized males and females, because there is always a variation in a population so that both smaller and bigger males/females are present. Therefore, the SD should be considerably longer than the minimal distance to the external pore, otherwise, some males will not be able for sperm transfer.

The range of variability in length of the female structures is 1165 – 1205 µm. The same for male spermatodactyls is 1112 – 1285 (N= 9 and 11 respectively). This range is relatively small.

3) The spiral shape could have independently evolved in both sexes because of parsimony reasons. It is much easier to live with a compact spiral structure than with a long linear SD. Additionally, a spiral form is a universal way to make a structure mechanically stronger, which is especially important for insert structures and different internal tubular structures.

Interesting idea, but it does not address the question of why females should have such a structure (the far majority of mesostigmatid females do not), or why the female structure should match the male spermatodactyl structure. The question remains: why structures that are this complex, and why have matching structures in males and females

4) The sperm pocket is situated very close to the external pore and it is not clear why the SD should be inserted entirely.

The reviewer is correct and we hypothesize based on other systems (vertebrates and invertebrates) that the space in coxa II and IV might be a sperm pocket. However there is not evidence to support that in this species. If the reviewers recommend we will remove any mention of this in the manuscript.

5) When the SD is being inserted it should change its coiled shape, therefore any real “matching” between male and female spiral shapes disappear.

The spermatodactyl can “unfold”, but it cannot assume any shape. It naturally assumed the coiled shape also seen in the female, but cannot be inserted while coiled. We suggested uncoiling to insert while re-coiling inside the female fitting with the structure inside the female. 

We want to stress that we cannot explain all observations made. We simply provide the best hypotheses we can come up with, given the data

The discovery that CLSM is a powerful tool for studying external and internal structures of mesostigs is a very strong point of the reviewed MS. Although this aspect is described quite well, some broader discussion, including comparison with results obtained by previous authors, would make the MS stronger. Perhaps, the authors could compare their results with papers by Valdecasas, Cramerik, Haug, Bolton, Chetverikov. Additionally, there is data in the literature (https://doi.org/10.11646/zootaxa.4066.3.4), suggesting that the green laser is the best for observing Dermanyssus (Mesostigmata) under CLSM. Remarkably, in this study the authors found the red laser to be more appropriate for mesostigs. This fact probably deserves a brief discussion.

We review the suggested information and made a stronger discussion about the use of different lasers in Mesostigmata.

I feel the structure of some parts of the MS needs revisions. Some paragraphs could be transferred from the Results / Discussion to the Introduction (e.g. general data on the usefulness of LTSEM/CLSM – p.10, lines 56-65, and “initial assumption” on CLSM based on the data from literature – p. 12, lines 139-141).

This is a good point, we restructure the paper and transferred some paragraphs to different section.

Besides, some new unexpected data/materials on the mites from Amazonia are mentioned in the Discussion (p. 17, lines 196-201) impeding a clear understanding of the text. 

We tried to clarified it.

The figure captions, figure numbers, and arrows in the figures need careful check DONE. There are also several repetitions (e.g. the phrase “The left internal spiral structure has a clockwise direction and the right has a counterclockwise direction” can be found several times in the text).

The direction notes are critical. The fact that the left spermatodactyl coils counterclockwise, while the left female structure coils clockwise suggests 1) that the two are not just copies of similar structures (at some level), and 2) that the two match (given the mating position in Dermanyssina). 

In general, I think that the MS needs careful reconsideration, partial rewriting, and restructuring and suggest “minor revision”.

Philipp E. Chetverikov

Saint-Petersburg

02/11/2020

Reviewer #2: The manuscript by Cómbita-Heredia et al describes the genital morphology of a mesostigmatid mite using different microscopic techniques. By doing so, the authors also compared different techniques focussing especially on the suitability of CLSM for the study of cuticular structures. I enjoyed reading the manuscript and also think it will add value to the acarological research community.

The manuscript is well written, but I feel that it would benefit from restructuring to better separate the methodological part from the part about genitalic morphology and evolution. This is especially true for the figures. For example, on p.5 the authors describe the different results using different mounting media. They refer to supplementary figures and also the very useful table. I suggest to present the results of their methodological survey in a single comparative plate (besides the supplementary material) that readers can see the pro and cons of the different approaches immediately. I have one more comment to the section about the LT-SEM in the results section. The authors claim that color can be added to emphasize critical features. I completely agree with it, but was wondering why the authors didn´t do it. Instead of highlighting critical features the authors applied color to give the impression of a natural look of the structures. For example, the application of color in Fig 5 is completely misleading and does not add any value but rather confuses here. Especially the dark brown makes it hard to see any details and thus I strongly suggest to remove the color in Fig 5! In general, I would suggest to highlight only structures of interest (one of many examples: Labarque et al 2017, Zool J Linn Soc 181:308-341).

The reviewer is correct regarding that the colorized in this case is misleading, we change to the original image without colors 

As mentioned above, I strongly suggest to combine figures. For example, Fig. 1-3 can be certainly merged as each of them shows similar aspects. In this way the reader focusses only on one comparative figure showing the morphology of the spermatodactyl. Alternatively, the authors could consider to use Fig. 1 rather for a plate adressing the CLSM methodology and only merge Fig 2-3 DONE. Moreover, I was surprised about Fig. 7. I would have expected to see the interactive models of the spermatodactyl and secondary insemination system together with the respective figures, as e.g. 3D spermatodactyl combined with Figs. 2 and 3. Finally, the authors should also consider to move Figure 9 to the other figures of the spermatodactyl and combine it with Figure 4.

The reviewer is correct and some images were merged and figure 8 was removed . Figure 9 (which is now figure 8) was keep on the same place because it depicts the internal musculature and supports Video 1 on the movement of the spermatodactyl

Regarding the 3D models I would strongly suggest to invest some more time into them DONE. Why the authors did not use the chance to segment the different parts to provide a fully interactive 3D pdf model (one of many examples: Bicknell et al 2018, PLOS one, https://doi.org/10.1371/journal.pone.0191400, Fig. S1). Since the authors focus also on the methodology and since it is the first time applying CLSM/3D methods to mesostigmatid mites, the author could set some standards and provide a workflow which can be applied in future studies using the same methodology. In doing so, the authors could explore also semi-automatic segmentation tools like Biomedisa (https://biomedisa.de/;
https://www.nature.com/articles/s41467-020-19303-w), which can enhance such workflow.

The reviewer is correct in that it would be better to have the images segmented. The authors are aware of tools like Biomedisa because they are currently working with Micro CT scans. We explore this segmentation techniques but due to the nature of the 3D data (confocal microscopy) it was very difficult to get rid of out of focus information to make the automatic algorithm to recognize the exact structures. Therefore, after a lot of time exploring options to segment different structure, we decided that Drihsti paint was a better tool for segmenting confocal data. 

I find the title misleading. You did not test whether size, shape and directions matter. Moreover, using a single species you can hypothese a genital coevolution, but I would be more cautious and suggest to change the title.

The reviewer is correct we change the title and overall in the manuscript we give more emphasis in the matching of genital structure in mites and the microscopy techniques than in the evolutionary part of the paper. 

More minor comments:

- p.6, first paragraph Discussion, last sentence: What does "standard" mean? Please explain and explicitely describe what you mean here DONE!

- p.8, line 164ff: It reads that the system described here cannot be compared with similar systems in invertebrates but rather with vertebrates. I would rewrite this part as there are numerous example of invertebrates considering sexual conflict and genitalic evolution (one example: Kuntner et al. 2009, Evolution 63:1451-1463). DONE

- p.11, line 194ff: Please provide a figure showing the intra- and interspecific size difference fo the spermatodactyl! Did you correct the size of the spermatodactyl against body size?

We are currently working of the revision of the genus Megalolaelaps (currently 9 species) and the interspecific variation will be discussed there. Spermatodactyl was not corrected against body size but we will do it for the revision.

- p.12, line 232: The figure number is missing DONE.

- I was wondering whether the authors consider investigating a mating pair in copula by using MicroCT? In this way you can visualize the interaction of male and female genitalia during coupling in a direct way. I guess it would be worth adressing as interested readers might wonder about it.

The reviewer is correct and being able to capture matting with MicroCT will be ideal. This is something we have consider and explored previously. However, we have done experiments in laboratory to get this mite species to mate. They did mate successfully but due to their biology it was not possible to record or observe when and where this species mate. They are associated to dung beetles and our experiments suggest that they mate in the galleries of the beetle see (Combita-Heredia et al. 2020).

- Please label the different structures in Fig. 6 and explain in the caption where the arrows point to. DONE

---

## [Editor Report · Decision Letter 1]

8 Jul 2021

Size, shape, and direction matters: matching secondary genital structures in male and female mites using multiple microscopy techniques and 3D modeling

PONE-D-20-28641R1

Dear Dr. Combita-Heredia,

We’re pleased to inform you that your manuscript has been judged scientifically suitable for publication and will be formally accepted for publication once it meets all outstanding technical requirements.

Kind regards,

Matjaž Kuntner

Academic Editor

PLOS ONE
---

## [Editor Report · Acceptance letter]

5 Aug 2021

PONE-D-20-28641R1 

Size, shape, and direction matters: matching secondary genital structures in male and female mites using multiple microscopy techniques and 3D modeling 

Dear Dr. Cómbita-Heredia:

I'm pleased to inform you that your manuscript has been deemed suitable for publication in PLOS ONE. Congratulations! Your manuscript is now with our production department. 

Kind regards, 

on behalf of

Dr. Matjaž Kuntner 

Academic Editor

PLOS ONE